# Integrating Cellular Automata with Unsupervised Deep-Learning Algorithms: A Case Study of Urban-Sprawl Simulation in the Jingjintang Urban Agglomeration, China

**Cong Ou [1], Jianyu Yang [1,\*], Zhenrong Du [1], Xin Zhang [2] and Dehai Zhu [1]**

[1] College of Land Science and Technology, China Agriclutural University, No.17 Tsing Hua East Road, Haidian District, Beijing 100083, China; oucong@cau.edu.cn (C.O); duzhenrong@cau.edu.cn (Z.D.); zhudehai@cau.edu.cn (D.Z.)

[2] Institute of Electronics, China Academy of Sciences, No.19 North Fourth Ring West Road, Haidian District, Beijing 100190, China; xzhang@mail.ie.ac.cn

\* Correspondence: ycjyyang@cau.edu.cn

**Abstract:** An effective simulation of the urban sprawl in an urban agglomeration is conducive to making regional policies. Previous studies verified the effectiveness of the cellular-automata (CA) model in simulating urban sprawl, and emphasized that the definition of transition rules is the key to the construction of the CA model. However, existing simulation models based on CA are limited in defining complex transition rules. The aim of this study was to investigate the capability of two unsupervised deep-learning algorithms (deep-belief networks, DBN) and stacked denoising autoencoders (SDA) to define transition rules in order to obtain more accurate simulated results. Choosing the Beijing–Tianjin–Tangshan urban agglomeration as the study area, two proposed models (DBN–CA and SDA–CA) were implemented in this area for simulating its urban sprawl during 2000–2010. Additionally, two traditional machine-learning-based CA models were built for comparative experiments. The implementation results demonstrated that integrating CA with unsupervised deep-learning algorithms is more suitable and accurate than traditional machine-learning algorithms on both the cell level and pattern level. Meanwhile, compared with the DBN–CA, the SDA–CA model had better accuracy in both aspects. Therefore, the unsupervised deep-learning-based CA model, especially SDA–CA, is a novel approach for simulating urban sprawl and also potentially for other complex geographical phenomena.

**Keywords:** urban-sprawl simulation; cellular automata; transition rules; unsupervised deep-learning algorithms; deep-belief networks; stacked denoising autoencoders

---

## 1. Introduction

Urban areas have been dominant players in the world's socioeconomic, political, cultural, and environmental spheres, and the global shift from rural to urban living has been a defining trend. In the mean time, as the most significant land-use change processes, the urban sprawl has had an important impact on Earth's surface, ecosystem, and environmental sustainability, and is closely related with the life of almost half of the world's population [1]. In this context, urban-sprawl simulations have played a key role in understanding its spatial-evolution process and has became a powerful tool for supporting urban planning and sustainable development in urban agglomeration. In recent decades, several models were proposed to better understand and forecast urban-sprawl processes. Models evolved from early empirical statistical models or aspatially oriented models that are based on equations to reach static

status in the current spatial dynamic, to spatially oriented or integrated models that are based on the principle of spatial interaction. Comparing the two main models, the major difference is that the former models just predict population and land demand in the future from the perspective of economics or statistics, while the latter models further consider the spatial interaction in the process of urban sprawl. Among them, cellular automata (CA) are examples of mathematical systems constructed from many identical components, each simple, but together capable of complex behavior [2]. Because of the simplicity, flexibility, and intuitiveness of CA [3], it has been widely adopted as a typical spatial dynamic model to simulate urban sprawl.

When we use CA as a "bottom–up" model to simulate a complex geographical phenomenon like urban sprawl, the key step is how to define the transition rules of CA that determine the state conversion of geographical processes [4]. With many driving factors involved, there exists a nonlinear relationship between driving factors and geographical structure, and it is difficult to obtain appropriate transition rules. A variety of mathematical methods were used to obtain the appropriate transition rules of CA. Although a traditional method like multicriteria evaluation [5–7] is simple and its mechanism is clear, it is still not efficient and reliable because the determination of parameters has a certain subjectivity and randomness. It is also difficult to tackle a series of complex behaviors associated with natural systems. To overcome this problem, a series of machine-learning algorithms have been proposed, assuming that the historic geographical processes remain stable for a certain period in the future through local interaction between cells, such as logistic regression [8–10], artificial neural networks [11–14], support vector machines [15–17], decision trees [18,19], random forests [20,21], genetic algorithms [22,23], and swarm-intelligence algorithms [24–28]. Although the algorithms above show significant improvement in defining nonlinear transition rules for CA, there still remain many problems like overfitting, easily resulting in local optima and weak in global searching. Moreover, the geographical simulation of large-scale regions with fine-resolution units has become an inevitable trend [29], which further makes the implementation of big-sample-oriented CA simulations difficult with these methods.

In recent years, as a feature-learning method that can yield more nonlinear, more abstract representations [30], deep learning has become the dominant technique to learning more information from multivariable nonlinear systems in the era of Big Data. Compared to traditional machine-learning algorithms, deep-learning-based methods attempt to model high-level abstractions in data by using multiple processing layers with complex structures, resulting in better representations from the point of view of simplifying a learning task from input examples [31]. A deep architecture consists of feature-detector units arranged in multiple layers: lower layers detect simple features and feed into higher layers, which in turn detect more complex features [32]. In particular, two typical unsupervised deep-learning algorithms for training, deep-belief networks (DBN) [33] and stacked denoising autoencoders (SDA) [34], are regarded as a breakthrough in feature learning due to their effective training strategies. They are all based on a similar central idea: greedy layerwise unsupervised pretraining, followed by supervised fine-tuning [35]. Specifically, this idea provides a better approach to (pre)train each layer in turn, initially using a local unsupervised criterion [36] with the aim of learning to produce useful higher-level representations from lower-level-representation output of the previous layer, which leads to much better solutions in terms of generalization performance. Due to such characteristics, DBNs and SDAs were successfully implemented in many nonlinear systems like dimensionality reduction [37–39], time-series forecasting [40–42], acoustic modeling [43–45], and digit recognition [46–48]. Therefore, we think the above-mentioned algorithms also have the potential to be applied in urban-sprawl simulations.

In summary, urban-sprawl simulation is crucial to understand and assess the sustainable development of urban land use changes. In defining the transition rules of CA, current studies tend to introduce many varied machine-learning algorithms to tackle the difficulty in automatically extracting nonlinear relationship between driving factors and spatial dynamic processes. However, most traditional machine-learning-based CA models are small-sample-problem-oriented approaches by using feature

mapping, which is not suitable for representation learning in the era of Big Data. Therefore, we applied two typical unsupervised deep-learning algorithms (DBN and SDA) for representation learning from a complex geographical phenomenon like urban sprawl in this study. In addition, we noticed that there were a few studies [49–51] that applied deep learning to modeling geographical phenomena, but only when comparing the proposed deep-learning algorithm with traditional machine-learning algorithms, without a comparison of a series of similar deep-learning algorithms in the same case. No relevant models have been applied to large-scale cases such as urban agglomerations. Therefore, the purpose of this study is to incorporate unsupervised deep-learning algorithms into CA, making it possible to more accurately simulate urban sprawl. First, we integrated DBN and SDA, two typical unsupervised deep-learning algorithms, into the CA. Then, we selected the Beijing–Tianjin–Tangshan (JJT) urban agglomeration to test the proposed models because of its dramatic land-use changes and rapid urbanization in the past two decades. Finally, we compared the performance of the unsupervised deep-learning algorithms between themselves, and with the obtained results under traditional machine-learning algorithms like an artificial neural network (ANN) and logistic regression (LR) by using the figure of merit (FoM), the hit-miss-false alarm approach, and a series of landscape indices.

## 2. Methodology

### 2.1. DBN

A DBN is a typical multilayer generative neural network, and it exhibits particular strength for learning representations of large amounts of data with multiple levels of abstraction [52]. As shown in Figure 1a, a typical DBN architecture contains an unsupervised learning subpart by using Restricted Boltzmann Machines (RBMs), which is trained in a greedy manner, followed by a supervised fine-tuning subpart in the top level for prediction. The basic idea of DBN is to use an unsupervised learning method to train each RBM layer by layer, and finally fine-tune the whole network with supervised learning.

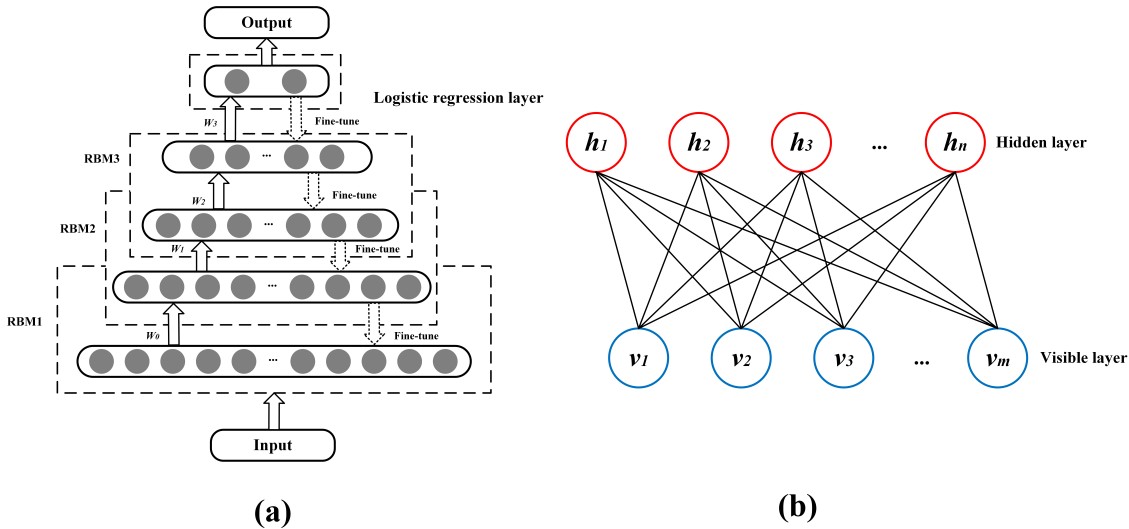

**Figure 1.** Architecture of (**a**) Deep Belief Networks (DBNs) and (**b**) Restricted Boltzmann Machines (RBMs).

As the basic DBN component, the RBM is in essence a stochastic neural network and an energy-based model. It is generally regarded as a kind of unsupervised learning algorithm to extract features from high-dimensional data by estimating their probability distribution. The structure of a typical RBM is shown in Figure 1b. It is a feed-forward graph network with two layers, the so-called visible layer (or input layer) of the $i$ dimension, representing observable data, and the hidden (or output) layer of the $j$ dimension, representing extracted features from observable data. Moreover, it allows full connection

within the different layers, but no connections within the same layer. The purpose of unsupervised learning by RBM is to continuously adjust the connection weight and bias parameters through Gibbs sampling, so as to minimize the error between $v'$ and $v$ after the restoration and reconstruction of implicit characteristic signal $h$. Given the difficulty of determining the step size for calculating the slope of a probability change, an algorithm called contrastive divergence (CD) [53] is proposed to speed up the RBM training process and maintain accuracy.

From Figure 1b, we can see that each unit of the visible layer has a symmetric connection weight with the corresponding units of the hidden layer. Weight matrix $W$(size: $m \times n$) encodes a statistical relationship by the joint distribution between the visible and the hidden layer, which can be mathematically described as the following equations:

$$p(v,h) = \frac{e^{-E(v,h)}}{\sum_{v,h} e^{-E(v,h)}} \tag{1}$$

where $v$ is the visible vector, and $h$ is the hidden vector. If the visible units are binary-valued, energy function $E(v,h)$ of certain configuration can be defined as equation

$$E(v,h) = -\sum_{i=1}^{m} a_i v_i - \sum_{j=1}^{n} b_j h_j - \sum_{i=1}^{m}\sum_{j=1}^{n} h_i W_{ij} v_j \tag{2}$$

where $a_i, b_j$ are the biases of the visible and hidden layer, respectively, and $W_{ij}$ is the combined weights.

In summary, an RBM contains five parameters: $v$, $h$, $W$, $a$, and $b$. Except for $v$ being the visible (input) vector and $h$ the hidden (output) vector, $W$, $a$, and $b$ can be learned and updated by computing the extract gradient of the log probability of the training data. According to the CD algorithm, the updating rule for $W$, $a$, and $b$ is as follows:

$$\Delta W_{ij} = \varepsilon(< v_i h_j >_{data} - < v_i h_j >_{reconstruction}) \tag{3}$$

$$\Delta a_i = \varepsilon(< v_i >_{data} - < v_i >_{reconstruction}) \tag{4}$$

$$\Delta b_j = \varepsilon(< h_j >_{data} - < h_j >_{reconstruction}) \tag{5}$$

where $\varepsilon$ is the learning rate, $<>_{data}$ is the expectation of the observed data, and $<>_{reconstruction}$ is the expectation under the reconstructed visible and hidden units, respectively, at every iteration.

To overcome the limited learning ability of the single-layer network, the DBN is proposed as a multilayer learning structure that is built in the form of stacks using individual restricted Boltzmann machines. The DBN training process can be divided into two main stages as follows [54]:

- *Unsupervised pretraining stage*: (1) Train the bottom RBM with CD on training data $v$ being the visible units; (2) freeze weights matrix $W$ and bias $a$, $b$ of the first RBM, and the state of the hidden units is inferred and used as the input of the next higher RBM; (3) next higher RBM is stacked on top of the previous lower RBM after training; (4) iterate Steps 2 and 3 for the desired number of layers, each time upward propagating either samples or mean values.
- *Supervised fine-tuning stage*: After an unsupervised pretraining stage, all parameters need to be slightly adjusted in supervised manner until DBN loss function reaches its minimum. In this paper, a logistic regression layer periodically works in the top-level RBM during the supervised fine-tuning stage.

## 2.2. SDA

Similar to DBN, which consists of a series of RBMs, an SDA is established based on a series of denoising autoencoders (DAEs; see Figure 2a) and learns to use a deep network architecture in a layer-by-layer fashion. The key function of SDA is unsupervised pretraining; once each layer is

pretrained for feature selection and the extraction of input from the previous layer, the second phase of supervised fine-tuning can occur.

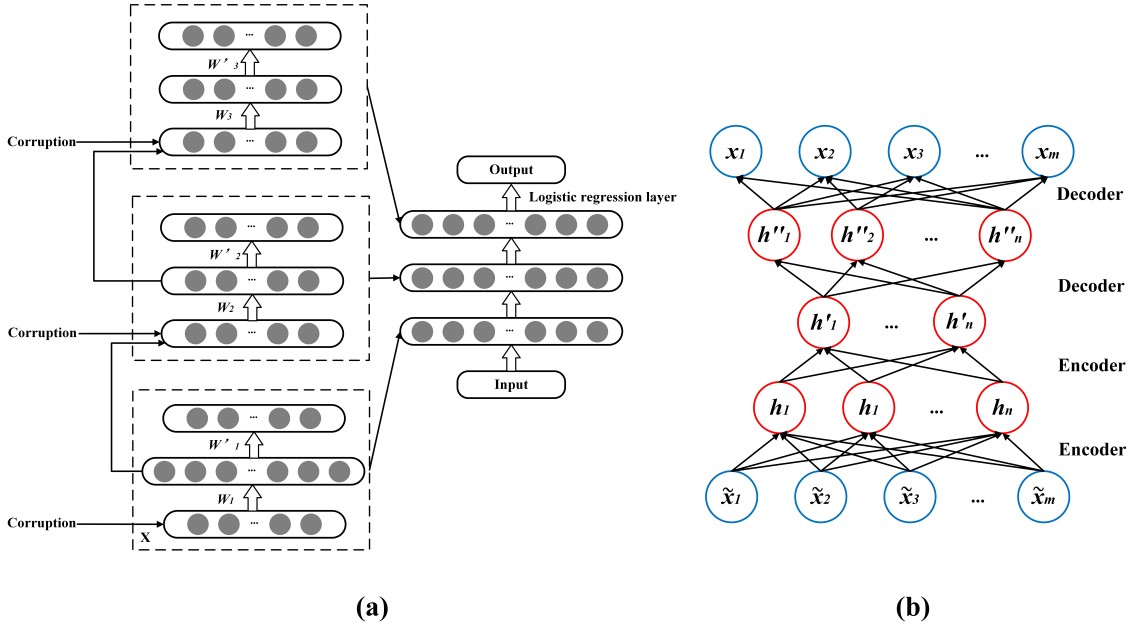

(a)　　　　　　　　　　　　　　　　　　　　　　　　(b)

**Figure 2.** Architecture of (**a**) Stacked Denoising Autoencoder (SDA) and (**b**) Denoising Autoencoder (DAE).

The autoencoder (AE) learns from a distributed representation (encoding) for a set of training data, and then reconstructs the data back to themselves from the encoder (decoding). However, it simply retains the information of the original input during the learning process, thereby failing to ensure that useful feature representation is extracted for the input data. To avoid this, the DAE adds some statistical noise to the original input and "corrupts" the original input, so that the autoencoder not only learns the characteristics of the original data, but also learns degraded features after being "corrupted", which greatly improved the overfitting problem and the generalization ability of the AE.

From Figure 2b, we can see that a DAE consists of an encoder, a decoder, and a hidden layer. It is trained to reconstruct a clean "repaired" input from a "corrupted" version. First, add statistical noise to original input **x** to obtain partially "corrupted" data $\widetilde{x}$ by a stochastic mapping:

$$\widetilde{x} \sim q_D(x'|x) \tag{6}$$

where $D$ represents the dataset. After being corrupted, the input to the encoder is $x'$ after the original data are corrupted, and hidden layer **y** is computed by

$$\mathbf{y} = f_\theta(\widetilde{\mathbf{x}}) = s(\mathbf{W}\widetilde{\mathbf{x}} + \mathbf{b}) \tag{7}$$

where $s$ represents a nonlinear mapping function; $W$ is the weight matrix; $b$ is the bias vector; and $\theta$ is the collection parameter for the mapping. Next, the reconstruction layer is generated by

$$\mathbf{z} = g_{\theta'}(\mathbf{y}) = s(\mathbf{W}'\mathbf{y} + \mathbf{b}) \tag{8}$$

where $W'$ is the inverse weight matrix; $b$ is the inverse bias vector; and $\theta'$ is the collection parameter for the inverse mapping. Finally, reconstruction loss $L$ measured by the squared in:

$$L(\mathbf{x}, \widetilde{\mathbf{x}}) = \|\mathbf{x} - \widetilde{\mathbf{x}}\|^2 \tag{9}$$

The training process of a DAE aims at minimizing the summation of squared errors over all training data. After minimizing the overall squared loss function, DAEs are trained.

In summary, an SDA is formed by stacking several DAEs through a layer-by-layer overlay mechanism similar to DBN. Its training mechanism is essentially a layer-by-layer greedy training mechanism that is also divided into two stages: unsupervised pretraining and supervised fine-tuning. The details are as follows:

- *Unsupervised pretrain stage*: (1): Train the bottom DAE through the above steps, and an encoder $f_\theta$ is obtained when the first layer is trained; (2) the feature representation vector is obtained by this encoder $f_\theta$ on the original input data and regarded as the hidden layer vector, which is used to obtain and train the encoder of the second SDA layer; (3) iterate Steps 1 and 2 for the desired number of SDA layers.
- *Supervised fine-tuning stage*: When the entire pretraining stage is over, the top layer is the final output layer. With this output as the base layer for logistic regression errors throughout the SDA structure, fine-tuning global parameters are adjusted.

*2.3. Proposed Geographical CA Model*

A standard CA consists of a cell space and a transition function that defines the space [55]. The basic CA components are the state of the cell, cell space, neighborhood, transition rules, and discrete time. So, a cellular automaton can be described by the following five-tuple model:

$$CA = \{S, L, N, R, T\} \tag{10}$$

Where $S$ is the state of the cell, all cells are mutually discrete, and a cell can only have one state at a certain time, which is taken from a finite set; $L$ represents the lattice, the cell that is distributed in the set of spatial sites, and the commonly used two-dimensional cellular automaton includes a triangular mesh, a square mesh, and a hexagon mesh; $N$ describes the spatial neighborhood of the cell, which is a collection of cells delimited by a certain shape around the cell, and it can affect the state of the cell at the next moment; $R$ represents the transition rules, which are the core of the cellular automaton, and it expresses the logical relationship of the simulated process and determines the result of a spatial change; Time interval $T$ is an integer value and may be equally spaced, and assuming that the initial moment is $T_0$, then the state of a cell at time $T_0 + T$ depends on the state of the cell and its neighbor cells at time.

Geographical CA includes geospatial-change information acquisition, global spatial-variable acquisition, transition-rule definition, model construction, and process simulation [11]. Change-information acquisition and global spatial-variable acquisition are usually done by GIS/RS tools. Model construction and process simulation are usually done by GIS tools or an open-source programming framework. Transition rules are a key part of building a model and directly determining the quality and effectiveness of a geographical CA. For an urban sprawl, the cell's state is guided by transition rules, which reflect the complexity of the environment. These rules act as a link between spatial patterns and the underlying spatial process [56]. This is deeply affected by a series of driving factors like the initial state of a cell, neighborhood conditions, global spatial variables, and some constraints on urban development. Therefore, an urban-sprawl simulation model can be formulated as follows:

$$S_{ij}^{T_0+T} = R(S_{ij}^{T_0}, N_{ij}^{T_0}, G_{ij}^{T_0}, C_{ij}^{T_0}) = \begin{cases} urban, & p_{ij}^{T_0} > p_{threshold} \\ nonurban, & p_{ij}^{T_0} \leq p_{threshold} \end{cases} \tag{11}$$

where $S_{ij}^{T_0}$ and $S_{ij}^{T_0+T}$ represent the state (urban/nonurban) of cell *ij* at time $T_0$ and $T_0 + T$, respectively. $p_{ij}^{T_0}$ is the probability of urban sprawl of cell *ij* at $T_0$, $p_{threshold}$ is the threshold value generally predefined in the range of [0,1]. $N_{ij}^{T_0}$ represents the state of neighboring cells. $G_{ij}^{T_0}$ represents global spatial

variables, such as socio-economic factors and natural environmental conditions. $C_{ij}^{T_0}$ are the constraint conditions for urban sprawl. $R$ are the urban-sprawl transition rules.

In addition, the various global spatial variables in the transition rules correspond to many weights that reflect the degree of contribution of different variables to the model. The setting of these weights has a great influence on the model simulation results. In practical implementations, it is difficult to effectively define the transition rules of these different categories of variables. Therefore, we used DBN and SDA to simplify this complex process with a training network to directly obtain the transition probability of every cell.

The proposed model consists of two parts (see Figure 3): The first part of the DBN (or SDA)-based CA model is to train the DBN (or SDA) to obtain the transition probability from a training dataset that is the sample of historic data and obtain a well-trained model. The simulation is cell-based, and each cell is composed of a set of attributes as inputs to the DBN (or SDA) after converting them in the range of $[0, 1]$. The attributes during the training phase can be expressed by:

$$X = [x_1, x_2, x_3, ..., x_n]^T \qquad (12)$$

where $x_i$ is the $i$th attribute and $T$ is transposition. As described above for DBN and SDA, the attribute inputs of these series of cellular automaton are directly used as the observed data for the unsupervised pretraining stage by using the DBN energy function (see Equation (2)) or the SDA loss function (see Equation (9)). Accordingly, each cell has its own state at the next time that can be expressed by:

$$Y = [y_1, y_2, y_3, ..., y_n]^T \qquad (13)$$

where $y_i$ is the $i$th state and $T$ is transposition, which are regarded as the outputs to slightly adjust all parameters of DBN (or SDA) for the supervised fine-tuning stage.

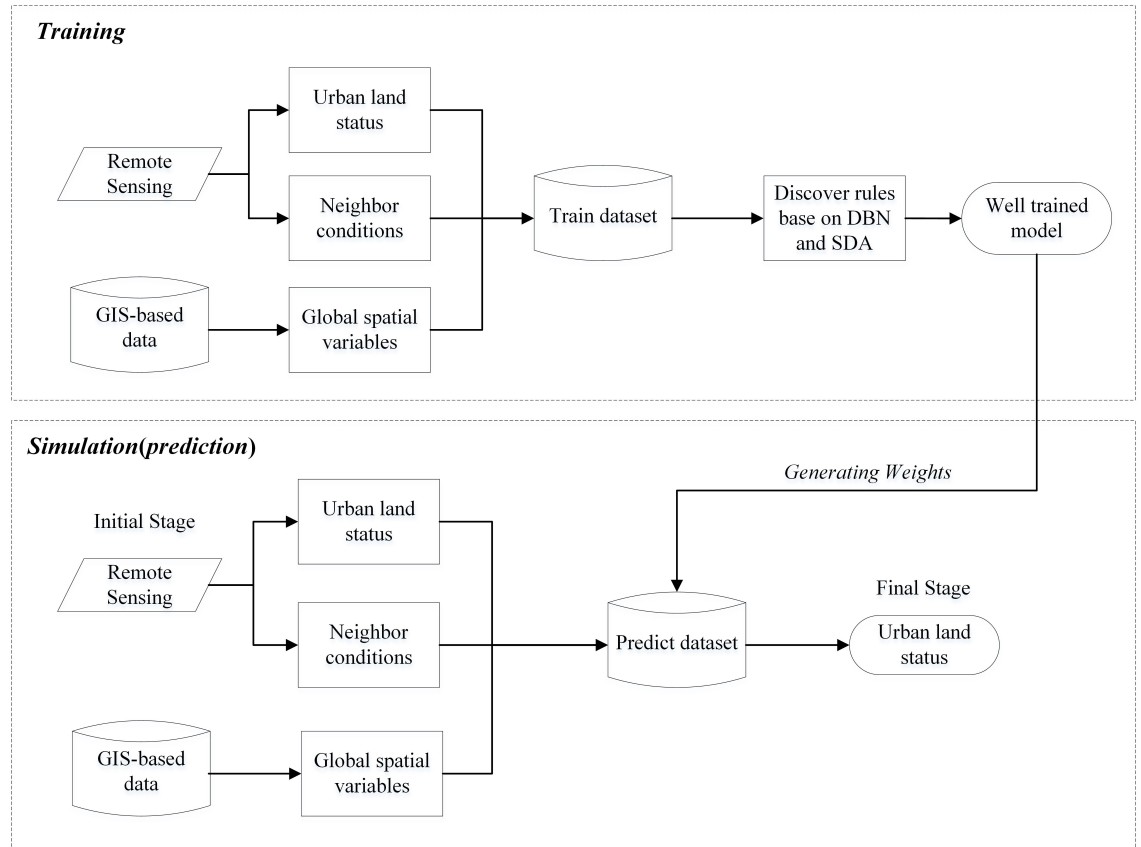

**Figure 3.** Flowchart of the proposed model.

The second part is to simulate (or predict) an urban sprawl by applying a prediction dataset to the well-trained model. Just like the training phase of the networks, the network input is the state of each cell itself, the state of the neighborhood, and other global spatial variables, and the output is the state of each cell at the next moment.

*2.4. Accuracy Assessment*

Ten indices were used as the criteria to evaluate the predictive performance of the proposed CA models in this study, which can be grouped into two types: criteria for assessing position accuracy at the cell level, and measures of pattern accuracy at the pattern level. At the cell level, we used FoM and the hit-miss-false alarm approach. Specifically, FoM is the ratio of the intersection of simulated and observed changes to the union of the simulated and observed changes [57], which can be calculated as follows:

$$FoM = \frac{\Delta_{simulated} \cap \Delta_{observed}}{\Delta_{simulated} \cup \Delta_{observed}} = \frac{B}{A + B + C + D} \tag{14}$$

where $\Delta_{simulated}$ is the numbers of changed cells by the simulated map compared to the initial map; $\Delta_{observed}$ is the numbers of changed cells by the observed map compared to the initial map; $A$ is the numbers of incorrect cells due to observed changes simulated as persistence; $B$ is the numbers of correct cells due to observed changes simulated as change; $C$ is the numbers of incorrect cells due to observed changes simulated as a wrong category; and $D$ is the numbers of incorrect cells due to observed persistence simulated as change.

The hit-miss-false alarm approach is a method used to quantify the goodness of fit of a land-change projection along a gradient of an explanatory variable [58], and it classified the pixels of a simulated map in this study as one of four types: correct due to observed urban sprawl predicted as urban sprawl (hits), error due to observed urban sprawl predicted as nonurban persistence (miss), error due to observed nonurban persistence predicted as urban sprawl (false alarms), and correct due to observed nonurban persistence predicted as nonurban persistence (correct rejection).

At the pattern level, a series of landscape indices, including the number of urban patches (NP), area-weighted mean patch fractal dimension (AWMPFD), edge density (ED), landscape shape index (LSI) and aggregation index (AI) were used in this study. Among them, NP, AWMPFD, and ED can be used to evaluate the fragmentation of the urban landscape, AWMPFD and LSI can be used to measure the complexity of the urban landscape, and AI can be used to assess the compactness of the urban landscape. The spatial urban form was measured based on a concise formula using the above-mentioned indices, which can be expressed as follows:

$$S_i = 1 - \frac{1}{n} \sum_{i=1}^{n} \Delta I_i \tag{15}$$

$$\Delta I_i = \frac{|I_{i,s} - I_{i,o}|}{I_{i,o}} \times 100\%, I = NP, AWMPFD, ED, LS, AI \tag{16}$$

where $I_{i,s}$ and $I_{i,o}$ are the $i$th landscape indices of the simulated and observed map, respectively, and $\Delta I_i$ is the normalized result of all landscape indices. $S_i$ is calculated by means of $\Delta I_i$, and can effectively reflect the similarity between the simulated and the observed map at the pattern level. $n$ is the number of landscape indices, set as 5 in this study since five types of landscape indices were used.

## 3. Proposed-Model Implementation

*3.1. Study Area*

The study area that we selected for implementing the proposed model is the JJT urban agglomeration, also known as Beijing–Tianjin–Tangshan, which is the national capital region of China and the largest urbanized region in northern China. The area is approximately 48,100 km$^2$

and contains a population of approximately 50 million. It contains two municipalities (Beijing and Tianjin) and two prefecture-level cities (Tangshan and Langfang; see Figure 4). Beijing is the capital of China, the world's third most populous city, and most populous capital city. Tianjin is the largest coastal metropolis and the third largest city in northern China. Beijing and Tianjin are regarded as the "dual core" of the JJT urban agglomeration and have accepted most migrants from other regions of northern China. The JJT urban agglomeration has been experiencing rapid urbanization since the implementation of reform and opening-up policy in 1978. During 2000–2010, the urban population and the GDP of the JJT urban agglomeration increased by about 5.34 million and RMB2383.01 billion (Yuan; approximately USD354.83 billion).

To construct a multiperiod CA model, this study adopted urban dynamics in 2000, 2005, and 2010. Specifically, the urban area in the study area occupies 4.13% of the total area in 2000, 5.53% in 2005 and 5.92% in 2010, generating an average sprawl area of 428.32 km$^2$ over this decade. With this rapid urban development, the land use of the JJT urban agglomeration has become very complex, leading to a series of environmental, economic, and urban-development issues. Although China has formulated the strictest land-use control and farmland-protection policies around the world, the influx of large populations inevitably leads to the continued sprawl of this area, and its land-use structure has experienced and will continue to experience dramatic changes. Therefore, an effective simulation of the urban sprawl in such an area can assist future urban planning and infrastructure construction, and guide the development of cities in a reasonable trajectory.

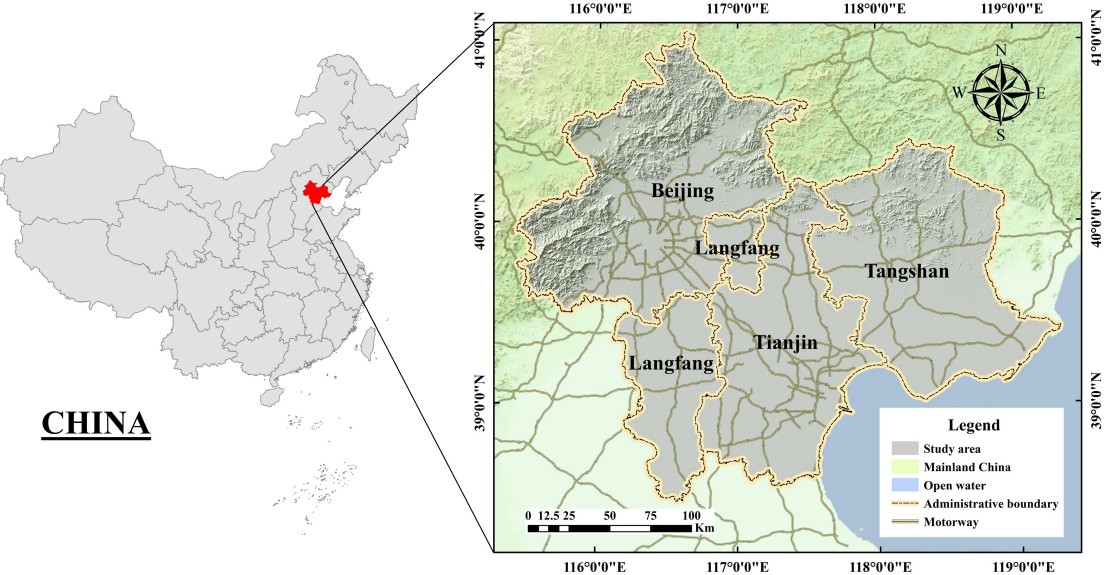

**Figure 4.** Location of Jingjintang urban agglomeration.

## 3.2. Data Preprocessing

### 3.2.1. Land-Use Data

A series of land-use maps (2000, 2005, and 2010) for the JJT were used in this study. All of these maps were produced from the classification of Landsat TM/ETM+ images (https://earthexplorer.usgs.gov/) with a resolution of 30 m, and have six land-use classes (urban land, cropland, grassland, forest, water, and others). They were used to derive past and current actual urban land, and detect urban sprawl (see Figure 5), which serve as the initial and final state of each cell during simulations in CA models. In the simulation phase, the target variable represents if the state of a nonurban cell remained (0), if a nonurban cell developed into an urban state (1), and if an urban cell remained (2) (here, we do not consider the case of an urban cell converting to a nonurban cell because this study only simulates the progress of the urban sprawl.)

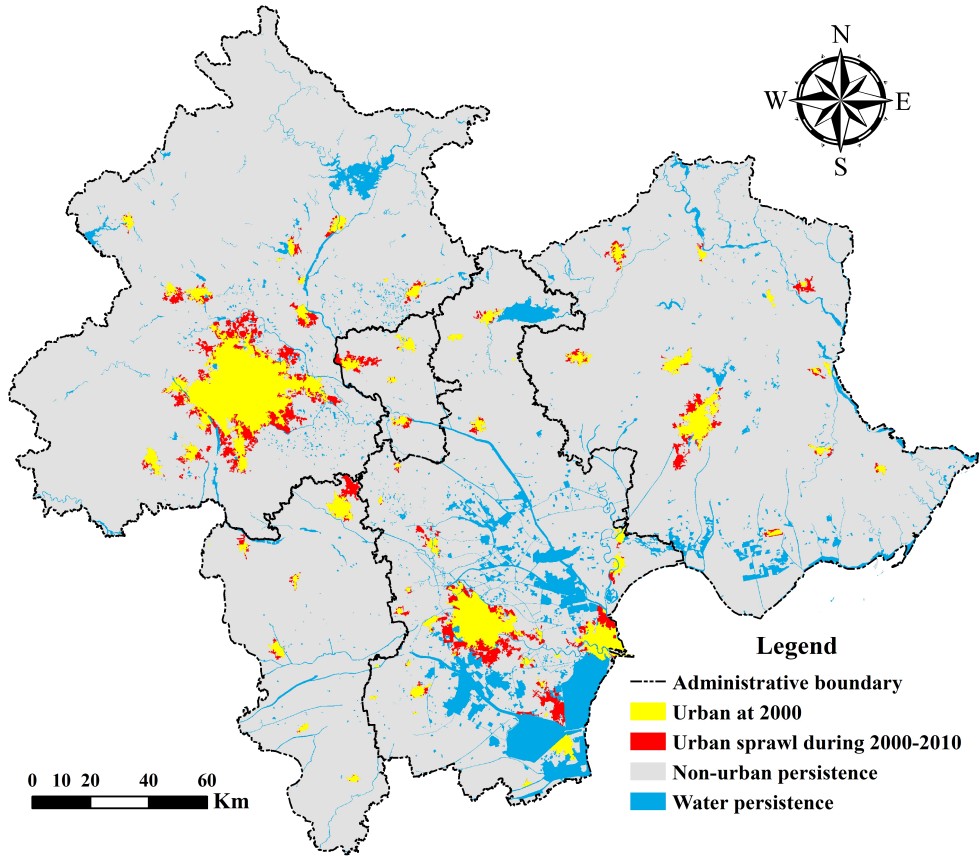

**Figure 5.** Urban sprawl in the study area from 2000 to 2010.

### 3.2.2. Neighborhood Conditions

The neighborhood conditions include both nonurban and urban neighborhoods, which is also important for the determination of an urban sprawl during the simulation. Although $3 \times 3$, $5 \times 5$, and $7 \times 7$ windows have been widely used in previous study, we performed a test in which we found that the $7 \times 7$ window performed best, and it may be because it contains more neighborhood interactions. Therefore, we chose the number of surrounding urban cells in $7 \times 7$ with the extended Moore neighborhood in the implementation. The states of the surrounding cells were derived from land-use maps using the FocalStatistics function of ArcPy.

### 3.2.3. Global Spatial Variables

Global spatial variables, as we mentioned above, have great influence on an urban sprawl. In this implementation, we chose a series of global spatial variables, including physical property and distance-based variables. The physical property mainly considered the influence of topography on urban sprawl, which included the digital elevation model (DEM) and slope. The DEM was derived from SRTM data (https://earthexplorer.usgs.gov/), and the slope was calculated from the DEM using the Slope function of ArcPy. Distance-based variables include distance to the nearest airport, distance to the nearest city center, distance to the nearest county center, distance to the nearest town center, distance to the nearest railway station, distance to the nearest reservoir, distance to the nearest motorway, distance to the nearest railway, distance to the nearest river, and distance to the nearest trunkway. This study accounts for distance-based variables from the aspects of accessibility, government/consumer decisions, and living environment. Among them, the vector data for airport, city center, county center, and town center were derived from the National Geomatics Center of China (http://www.ngcc.cn/), and the vector data for reservoir, river, railway station, motorway, trunkway,

and railway were derived from OpenStreetMap (http://www.openstreetmap.org). All distance-based variables were calculated from the vector data using the EucDistance function of ArcPy (see Figure 6).

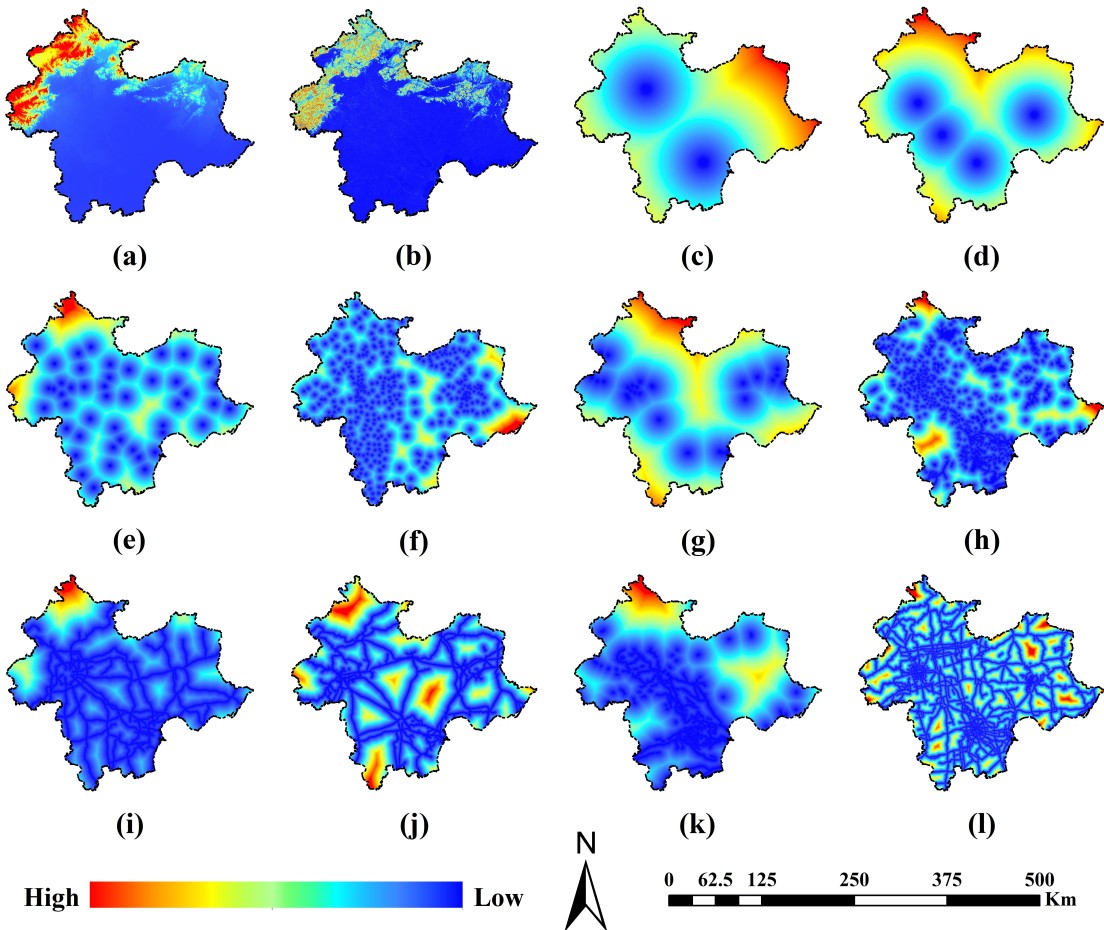

**Figure 6.** Spatial variables in metropolitan Jingjintang: (**a**) digital elevation model (DEM); (**b**) slope; (**c**) distance to airport; (**d**) distance to city center; (**e**) distance to county center; (**f**) distance to town center; (**g**) distance to railway station; (**h**) distance to reservoir; (**i**) distance to motorway; (**j**) distance to railway; (**k**) distance to river; (**l**) distance to trunkway.

As listed in Table 1, all data mentioned above were resampled to a spatial resolution of 100 × 100 m, with 3325 columns and 2910 rows, and restored them as the full dataset in CSV format to facilitate the proposed models. Then, using a systematic sampling approach, we selected 1,000,000 samples from the full dataset to serve as inputs to the proposed models for defining the CA transition rules. Because a large amount of data are available and efficient computing power in the era of Big Data, we did not have to compromise as much and could use a greater portion to train the model. Thus, samples were divided into three parts: 980,000 samples as training data used to fit DBN or SDA, 10,000 as valid data used to provide an unbiased evaluation of the DBN or SDA fit on the training data while tuning model hyperparameters, and 10,000 as test data used to provide an unbiased evaluation of a final model fit on the training data.

**Table 1.** List of spatial variables used in urban-sprawl simulation.

| Spatial Variable | Value Range |
|---|---|
| 1.*Target variable* | |
| Urban sprawl in 2000–2010 | Remained nonurban area: 0; converted to urban area: 1; remained urban area: 2 |
| 2.*Locational variables* | |
| Distance to airport | 0–175,594 m |
| Distance to city administrative center | 0–129,142 m |
| Distance to town administrative center | 0–47,261 m |
| Distance to county administrative center | 0–63,279 m |
| Distance to reservoir | 0–44,457 m |
| Distance to river | 0–88,911 m |
| 3.*Accessibility* | |
| Distance to railway station | 0–109,976 m |
| Distance to motorway | 0–70,052 m |
| Distance to trunkway | 0–21,979 m |
| Distance to railway | 0–33,278 m |
| 4.*Neighborhood condition* | |
| Urban cells in $7 \times 7$ neighborhood | 0–48 |
| 5.*Cell characteristic* | |
| Land use type | urban area: 1; nonurban area: 0 |
| Elevation | −52–2,224 m |
| Slope | 0-69° |

## 3.3. Model Design and Computational Environment

The entire implemented process, including data preprocessing, training of the proposed models, and simulation, is based on Python. Some basic scientific-computation libraries like Numpy, Pandas, and Rasterio were used for spatial-data organization, such as raster-data reading and their conversion to CSV. Arcpy was used for spatial-data analysis, such as neighborhood or distance analysis. Scikit-learn (https://scikit-learn.org/stable/) was used to develop the artificial-neural-network (ANN) and random-forest (RF) methods as method comparison.

In addition to the above scientific computation libraries, it is particularly important to emphasize that the computational framework for DBN and SDA in our study was based on Theano (http://deeplearning.net/software/theano/), which is an open-source project that has the following characteristics: (1) symbolic computation for tensors, (2) highly expressive, transparent GPU acceleration, (3) easily switches between CPU and GPU, and (4) easily integrates with Python ecosystems. When running DBN and SDA based on Theano, there are some parameters that need to be preset (see Table 2).

**Table 2.** Set parameters for DBN and SDA.

| Description | Symbol | DBN | SDA |
|---|---|---|---|
| Input dimension | *n_ins* | 11 | 11 |
| Output dimension | *n_out* | 3 | 3 |
| Intermediate layer size | *hidden_layer_sizes* | [30, 30, 30] | [30, 30, 30] |
| Number of epochs for pretraining | *pre-training_epochs* | 300 | 300 |
| Maximal number of iterations of running optimizer | *training_epochs* | 3000 | 3000 |
| Learning rate used in pretraining | *pretrain_lr* | 0.01 | 0.001 |
| Learning rate used in fine-tuning stage | *finetune_lr* | 0.1 | 0.01 |

## 3.4. Simulation Results and Comparisons

The DBN–CA and SDA–CA models proposed in this study were established through known data from 2000, 2005, and 2010. The initial year was 2000, and the 2000–2005 samples were used to calibrate the proposed models for discovering the transition rules; the simulation result of urban sprawl during 2000–2005 was conducted simultaneously. Next, the simulated urban-land status of 2005 was acquired to conduct the simulation result of the urban sprawl during 2005–2010 based on the calibrated models.

Land-use data in 2010 were regarded as the observed urban-land status to confirm the validation of the calibrated DBN–CA (or SDA–CA) model. Figures 7 and 8 show the consistency between the initial, observed, and simulated urban-land status and enlarged area of Beijing in 2005 and 2010 based on the four models mentioned above.

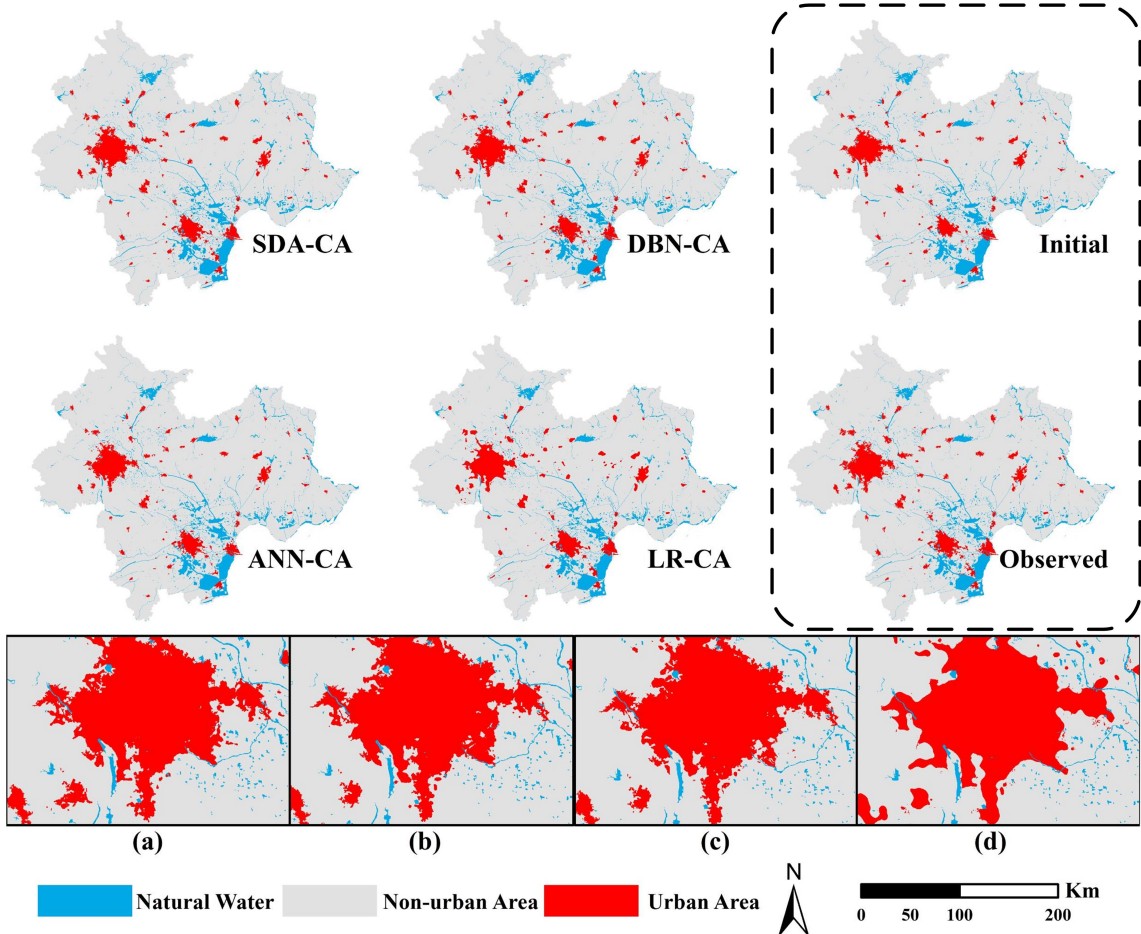

**Figure 7.** Comparison of simulated results by four models for 2005. (**a**) Enlarged area of SDA–cellular automata (CA), (**b**) enlarged area of DBN–CA, (**c**) enlarged area of ANN-CA, (**d**) enlarged area of LR-CA.

Additionally, FoM were used for more quantitative assessments of the proposed models at the cell level. As shown in Figure 9, SDA–CA obtain the best simulated results in both periods. Compared with DBN–CA, the FoM of SDA-CA increased by 9.8% in 2005 and 14.2% in 2010, which demonstrated that SDA is more suitable for discovering the transition rules for CA than DBN at cell level. Meanwhile, compared with the other two traditional machine-learning algorithms, the FoM of the SDA–CA increased by 16.6–31.6 in 2005, and 17.9–19.7% in 2010, which corroborated that SDA can learn more information from multivariable nonlinear systems than traditional machine-learning algorithms at cell level, thus obtaining better recognition of urban sprawl.

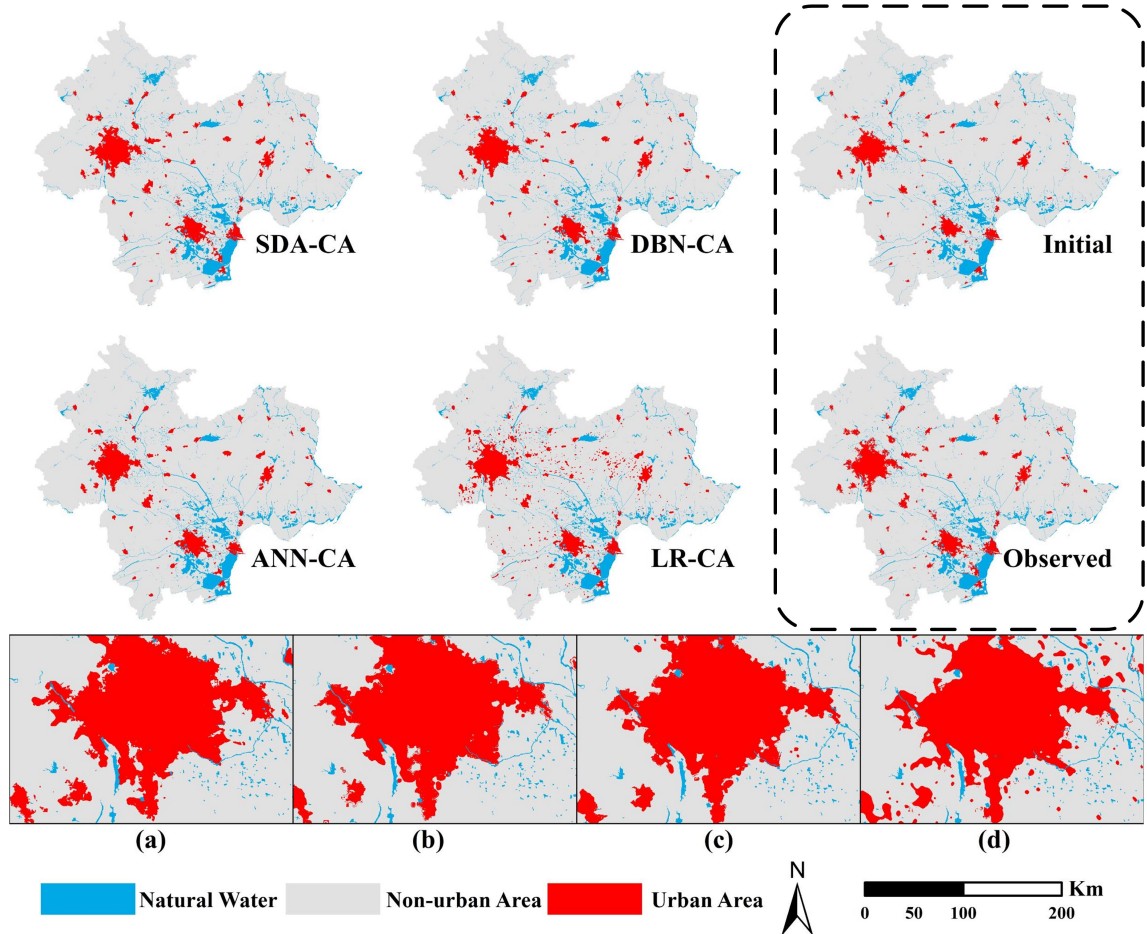

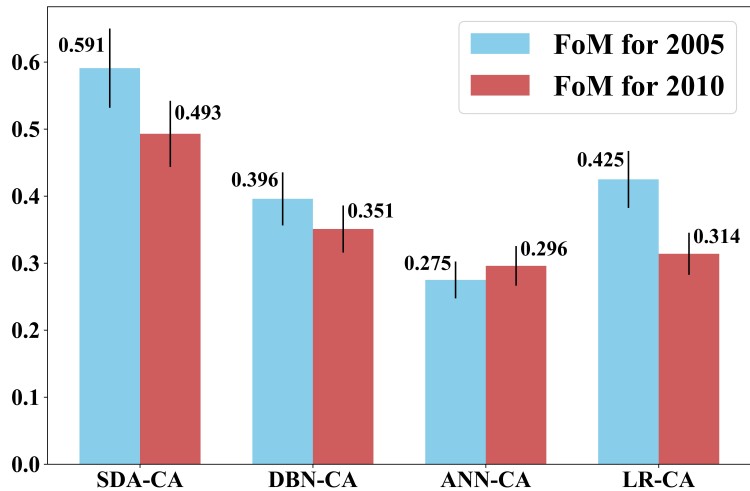

**Figure 8.** Comparison of simulated results by four models for 2010. (**a**) Enlarged area of SDA-CA, (**b**) enlarged area of DBN-CA, (**c**) enlarged area of ANN-CA, (**d**) enlarged area of LR-CA.

**Figure 9.** Figures of Merit (FoM) of simulated results by four models for 2005 and 2010.

In Table 3, we outline the FoM of the major city's simulation results in JJT by the four models for 2005 and 2010. In general, the minimum FoM of DBN–CA and SDA–CA in all cities for 2005 and 2010 reached 0.332, which is a reasonable result of the relevant research. The FoM of

unsupervised deep-learning algorithms are generally better than traditional machine-learning algorithms in every city. Comparing the FoM of four models, SDA–CA achieved the best result, and more specifically, the SDA–CA FoM of Beijing increased by 16.8–34.0% in 2005 and 12.5–19.1% in 2010, the SDA–CA FoM of Tianjin increased by 7.6–30.1% in 2005 and 8.3–21.0% in 2010, and the SDA-CA FoM of Tangshan increased by 21.2–26.4% in 2005 and 10.0–23.3% in 2010. These results demonstrate that the implementation effect of the model varied in different years or in different cities. From the perspective of development trends of various cities, taking the SDA-CA FoM as an example, Beijing and Tianjin, which are the municipality of China and a metropolis in northern China that developed at a high speed due to the effects of population aggregation, generally have a higher FoM of around 0.4–9.1% than Tangshan. These results illustrate that areas with a fierce urban sprawl are often able to achieve better simulation results because they provide more samples of the urban sprawl for training the models.

**Table 3.** FoM of major city's simulation results in JJT by four models for 2005 and 2010.

|  | SDA-CA | DBN-CA | ANN-CA | LR-CA |
|---|---|---|---|---|
| *2005* | | | | |
| Beijing | 0.583 | 0.385 | 0.243 | 0.415 |
| Tianjin | 0.581 | 0.403 | 0.280 | 0.505 |
| Tangshan | 0.579 | 0.367 | 0.320 | 0.315 |
| *2010* | | | | |
| Beijing | 0.468 | 0.343 | 0.277 | 0.294 |
| Tianjin | 0.523 | 0.364 | 0.313 | 0.440 |
| Tangshan | 0.432 | 0.332 | 0.299 | 0.199 |

In addition to using FoM to compare the simulation results of the four models, we also introduced the hit-miss-false alarm approach to measure the similarity between the simulated map and the observed map at the cell level. Four different types of results by four models for 2005 and 2010 are summarized in Table 4. Generally speaking, unsupervised deep-learning-based models have better performance than shallow machine-learning-based models in all four different types of results, and SDA–CA obtained the best results from the four models. Specifically, compared with DBN–CA, the numbers of SDA–CA hits increased by 6.11% in 2005 and 7.80% in 2010, the numbers of SDA-CA's miss decreased by 41.02% in 2005 and 39.21% in 2010, the numbers of SDA–CA's false alarm decreased by 22.48% in 2005 and 9.30% in 2010, and the numbers of SDA–CA's correct rejection increased by 0.08% in 2005 and 0.08% in 2010. Compared with the two other shallow machine-learning models, the numbers of SDA–CA's hits increased by 5.94–11.66% in 2005 and 8.41–10.34% in 2010, the numbers of SDA–CA's misses decreased by 43.04–55.77% in 2005 and 40.88–45.50% in 2010, the numbers of SDA–CA's false alarm decreased by 10.48% (but increased by 63.03% compared with ANN–CA) in 2005 and 22.50–49.56% in 2010, and the numbers of SDA–CA's correct rejection increased by 0.51% (but decreased by 0.03% compared with ANN–CA) in 2005 and 0.21–0.73% in 2010. These analysis results confirm the validity of the proposed models at cell level from another aspect.

**Table 4.** Hit-miss-false alarm of simulated results by four models for 2005 and 2010.

|  | SDA–CA | DBN–CA | ANN–CA | LR–CA |
|---|---|---|---|---|
| *2005* | | | | |
| Hit | 244528 | 230445 | 219001 | 230811 |
| Miss | 20247 | 34330 | 45774 | 35547 |
| False alarm | 12555 | 16195 | 11364 | 33964 |
| Correcct rejection | 4514091 | 4510451 | 4515282 | 4491099 |
| *2010* | | | | |
| Hit | 255148 | 236676 | 231238 | 235346 |
| Miss | 28638 | 47110 | 52548 | 48440 |
| False alarm | 32785 | 36148 | 42303 | 64997 |
| Correcct rejection | 4474850 | 4471487 | 4465332 | 4442638 |

To further demonstrate the results of our proposed models, here we enlarge the simulation results of SDA–CA to three major cities (Beijing, Tianjin, and Tangshan) in Figure 10. Visual inspection suggests that the hit area mainly concentrated on the vicinity of the initial urban area, the miss area mainly focused on where there is a sudden change in urban sprawl, and false alarm exists in urban fringe and nonurban areas. So, we can infer that the proposed model is more suitable for urban sprawling than a leap-forward development of urban areas.

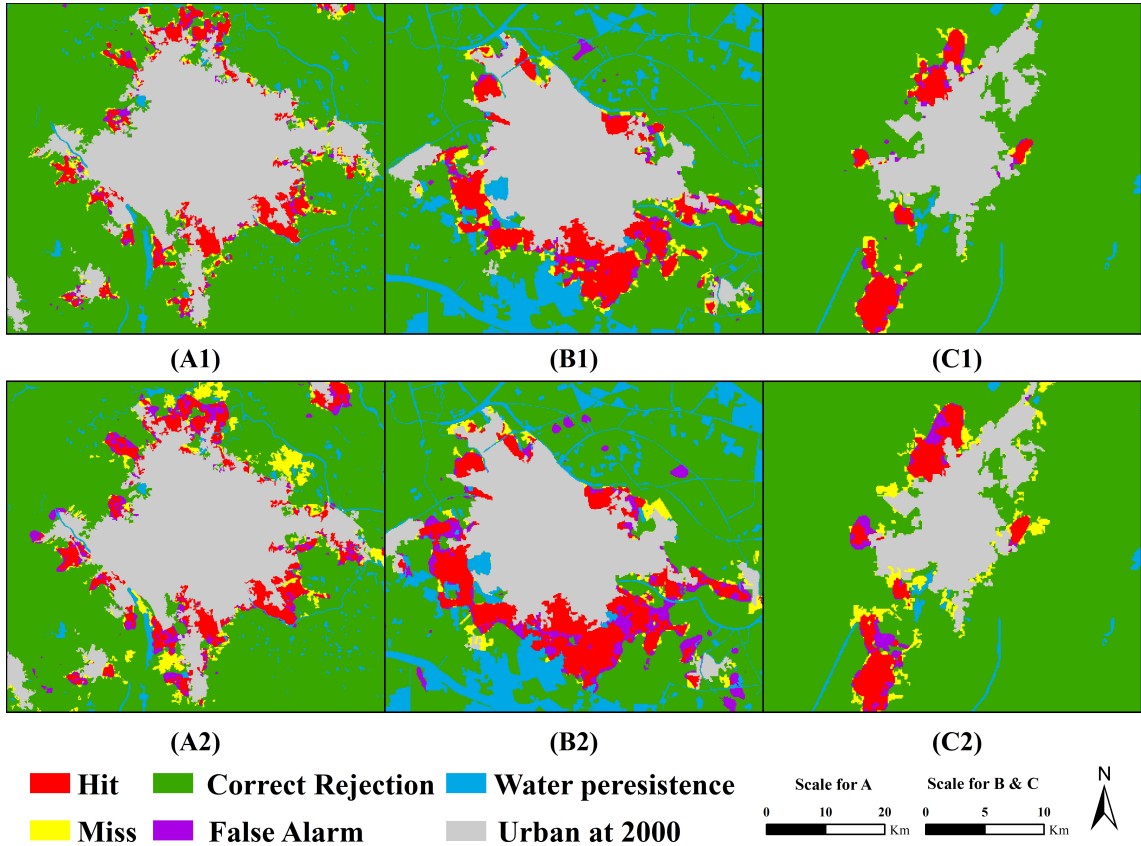

**Figure 10.** Comparison of enlarged areas in various cities based on detection theory. (**A1–2**) Enlarged area of Beijing in 2005 and 2010, (**B1–2**) enlarged area of Tianjin in 2005 and 2010, (**B1–2**) enlarged area of Tangshan in 2005 and 2010.

At pattern level, Table 5 lists a series of landscape indices for four different models in 2005 and 2010 that can distinguish the differences between the simulated and observed maps. In general, SDA–CA had the best performance in pattern similarity. As seen from the table, In the simulation results for 2005, the $S_i$ between the simulated map of SDA–CA and the observed map is 88.37%, with an NP value difference of merely 37, an ED value difference of merely 0.164, an LSI value difference of merely 3.504, an AWMPFD value difference of merely 0.016, and an AI value difference of merely 0.627. Compared with DBN–CA, the $S_i$ of SDA–CA increased by 3.29% in 2005 and 2.05% in 2010, which demonstrated that SDA is more suitable for urban sprawl than DBN at the pattern level. Moreover, compared with the two other traditional machine-learning models, the $S_i$ of SDA–CA increased by 7.64–38.08% in 2005 and 9.5–11.75% in 2010, which demonstrated that SDA is more suitable for urban sprawl than DBN at pattern level. On the whole, these results indicate the capability of SDA–CA for simulating the urban sprawl, and the SDA–CA model behaves well during the simulation at pattern level.

**Table 5.** Comparison of observed and simulated values of landscape indices.

|        | NP  | ED    | LSI    | AWMPFD | AI     | S_i    |
|--------|-----|-------|--------|--------|--------|--------|
| *2005* |     |       |        |        |        |        |
| observed | 142 | 1.019 | 23.889 | 1.175  | 95.539 | -      |
| SDA-CA | 179 | 0.855 | 20.385 | 1.159  | 96.166 | 88.37% |
| DBN-CA | 193 | 0.815 | 19.842 | 1.154  | 96.196 | 85.08% |
| ANN-CA | 229 | 0.818 | 20.614 | 1.155  | 95.905 | 80.73% |
| LR-CA  | 365 | 0.591 | 13.801 | 1.089  | 97.513 | 50.29% |
| *2010* |     |       |        |        |        |        |
| observed | 179 | 1.174 | 26.579 | 1.186  | 95.187 | -      |
| SDA-CA | 191 | 0.831 | 18.725 | 1.145  | 96.688 | 86.21% |
| DBN-CA | 229 | 0.884 | 20.384 | 1.148  | 96.281 | 84.16% |
| ANN-CA | 215 | 0.635 | 14.613 | 1.122  | 97.390 | 76.71% |
| LR-CA  | 361 | 1.080 | 23.705 | 1.101  | 95.846 | 74.46% |

## 4. Discussion and Conclusions

The urban sprawl has many negative consequences for residents and the environment, and how to accurately simulate this process for future planning has been a hot research topic of geographical simulations. Prior works documented the effectiveness of the CA model in simulating urban sprawls [3,9,11] and introduced a series of algorithms into the CA model to more efficiently discover its transition rules. However, these CA-based models either have remaining problems like overfitting, or have resulted in a local optimum. In addition, from the perspective of the development of machine learning, traditional shallow machine-learning algorithms can no longer adapt to current big-sample-oriented learning ideas in the era of Big Data. Therefore, we introduced two unsupervised deep-learning algorithms that achieved great power and flexibility in representation learning, for defining CA transition rules, and testing the effectiveness of the proposed models in an urban-sprawl simulation.

To test the effectiveness of the proposed models in different cities, we selected the JJT urban agglomeration as the study area and simulated its urban sprawl in 2000–2010. First, land-use data, neighborhood conditions, and global spatial variables in 2000, 2005, and 2010 were used as the full dataset for model calibration and validation. Then, based on the sample data from 2000 to 2005, two calibrated CA models were constructed by using DBN and SDA. Last, by using the two calibrated CA models, two simulated results of JJT in 2010 were conducted. Model validation was implemented by assessing consistency between the simulated and observed maps and comparing with two traditional CA models (ANN–CA and LR–CA models). As shown by a series of model accuracy assessments, the SDA–CA model obtained the best simulation results at both cell level and pattern level. Meanwhile, compared with other traditional machine-learning-based CA models, the DBN–CA model also had better performance in most aspects. These findings demonstrated that unsupervised deep-learning algorithms are well suitable for discovering CA transition rules, and the performance of the SDA–CA model may be better than the DBN–CA model for simulating complex geographical systems.

Most notably, this is the first study to our knowledge to investigate the capability of two typical unsupervised deep-learning algorithms for defining the transition rules of CA models in the same case. Our study provided a systematic comparison between unsupervised deep-learning-based CA models and traditional machine-learning-based CA models in a real urban-sprawl case. However, some limitations are worth noting. Although simulated results of the proposed models showed stronger capability in feature learning from urban-land-change data than traditional models, the relationship between an urban-sprawl state and driver factors (such as the neighborhood condition and global spatial variables) is still ambiguous. Future work should therefore not only introduce more deep-learning algorithms, such as RNN or GAN, to enhance the feature-learning ability of the CA model, but also needs to consider integrating the CA model with reinforcement learning [59] to improve simulation interpretability.

**Author Contributions:** C.O. and J.Y. conceived the study design, developed the models, and drafted the manuscript. J.Y. provided the funding. C.O. and Z.D. was involved in data acquisition and analysis, and worked on aspects of experiment evaluation. X.Z. improved the conceptual framework and updated the manuscript. D.Z. was involved in the design and revision of the manuscript. All authors read and approved the final manuscript.

**Funding:** This research was funded by the Chinese Universities Scientific Fund "Monitoring the quantity and quality using remote sensing and intelligence analysis of total factor for cropland", grant number 2019TC117 and the Special Fund for Scientific Research on Public Causes, grant number 201511010-06.

**Acknowledgments:** The authors would like to thank Wanling Chen and Yiming Liu for their suggestions in improving both the structure and the details of this paper.

**Conflicts of Interest:** The authors declare no conflict of interest.

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
