# Peer review of "Integrating Cellular Automata with Unsupervised Deep-Learning Algorithms: A Case Study of Urban-Sprawl Simulation in the Jingjintang Urban Agglomeration, China"

_sustainability, doi:10.3390/su11092464_

Round 1
Reviewer 1 Report
The followings are the comments regarding the study.
1. Line 1-2; Line 22-24. Simulation of urban sprawl has been regarded as a hot issue last decades. It might be great if authors can improve the contribution and importance of this study.
2. Line 24-28. As mentioned before, land use simulation is not a new topic, and there are many varied approaches applied in the simulation including Markov chain, what if, and so on. Please do a comparative analysis on various measurements.
3. Line 72-78. The last paragraph in introduction should be a conclusion of the overall contribution, importance, necessity, and method. Strongly suggest authors revise it.
4. Line 85. I am not familiar with DBN, and I am curious about the relationship between DBN and RBM. What is the role of RBM in this study?
5. Line 120. What is DAEs, and what is the relationship between DBN and DAEs?
6. Line 220. The more details regarding the study area would be great for the readers can understand the necessary of applying the framework on such rapid development urban areas. Therefore, more social and economic description would be help the readers understand.
7. Line 248. What is the neighborhood conditions? Why choose urban cells in 7 X 7? Will it different if base on different spatial scale unit?
8. Table 1. Why authors use the spatial variables including airport, city administrative center, reservoir, river, railway station? There should be a section on the rationality of selected variables.
9. Line 354. The main contribution of this article is applying fancy measurement on urban sprawl simulation. However, it would be great if authors can build more connections on such relationship. Currently, the paper is lacking of research question and contribution.
10. English professional editing is necessary including
(1) Line 19-20. “In the meaning time…” should be “In the mean time..”
Author Response
Thank you for the comments which are highly insightful and enabled us to improve the quality of our manuscript. The point-by-point response please see the attached file.

Reviewer 2 Report
The paper shows the analysis of machine learning methods for determining the transition probabilities of cellular automata simulating the urban sprawl phenomena.
The configurations space of the cellular automaton should be better specified, from a mathematical point of view, for the process with the energy function given by eq.2
The index γ of eq.3,4,5 (probably indicating "visible") should be better explained.
It would be also interesting to show the urban sprawl predictions for the next 50 years using the proposed method.
Author Response

(The authors gave the same response as above.)

Reviewer 3 Report
As someone well-versed in ML and CA, I really appreciated the fresh ideas and perspective brought by this paper - I enjoyed the paper very much and look forward to further developments along the lines suggested in the final section. In particular, I would be very interested to i) know how much we can learn with a purely local model (in the "original" spirit of automata, Wolfram-style) vs how much global info contribute to the prediction accuracy; ii) know how many timestamps we can model (in the spirit of "extending" the memory of the automata) when making a decision on what state a given cell is in at T.
Only one minor question re: the "980,000 , 10,000, 10,000" data split. Compared to a "normal" 70,20,10 split this is pretty aggressive: any reason to do this?
Author Response
Thank you for the comments which are highly insightful and enabled us to improve the quality of our manuscript. The point-by-point response please see the attached file.Thank you for the comments which are highly insightful and enabled us to improve the quality of our manuscript. The point-by-point response please see the attached file.

Round 2
Reviewer 1 Report
No comments.